# Does Internet Use Affect Medical Decisions among Older Adults in China? Evidence from CHARLS

**DOI:** 10.3390/healthcare10010060

**Published:** 2021-12-29

**Authors:** Gan Li, Chuanfeng Han, Pihui Liu

**Affiliations:** 1School of Economics and Management, Tongji University, 1239 Siping Road, Shanghai 200092, China; ligan@shutcm.edu.cn (G.L.); hancf@tongji.edu.cn (C.H.); 2Shanghai Museum of Traditional Chinese Medicine, Shanghai University of Traditional Chinese Medicine, Shanghai 201203, China; 3School of Management Engineering, Shandong Jianzhu University, Jinan 250101, China

**Keywords:** gatekeeping system, self-treatment, primary health institutions, CHARLS, PSM

## Abstract

Background: The rapid growth of the elderly population poses a huge challenge for people to access medical services. The key to get rid of the dilemma is for patients to go firstly to primary medical institutions. Existing studies have identified numerous factors that can affect patients’ health institution choice. However, we currently know little about the role of Internet use in the patients’ medical decisions. The objective of this study is to explore health-seeking behavior and institution choice under the background of the Internet era from the perspective of older adults, and to analyze whether the Internet could guide patients to the appropriate medical institution so as to accomplish hierarchical treatment. Methods: The dataset comprises 9416 people aged 45 or above from the China Health and Retirement Longitudinal Survey (CHARLS), which, through multistage cluster sampling, was conducted in 2011, 2013, and 2015. Logistic regression, PSM, and FE model are used to estimate the influence of Internet use on the health care decision-making behavior. Results: Internet use has a significant positive impact on the self-treatment of common diseases (β = 0.05, *p* < 0.05). In terms of medical institution choices, those who use Internet are more inclined to choose top-level hospitals than community health service institutions to treat common diseases (β = 0.06, *p* < 0.01). Conclusions: The Internet has lowered the obstacles to learning about common ailments, resulting in a substitution impact of self-treatment for hospital care. However, Internet use may aggravate older adults’ perception of the risk of disease, which exacerbates the tendency of going to higher-level medical institutions for medical treatment. The finding of the study is useful for further rational planning and utilization of the Internet in order to guide patients to appropriate medical institution, which helps to improve the efficiency of the overall medical and health services.

## 1. Introduction

With a rapidly ageing population, the Chronic non-infectious diseases are becoming more common, and an effective health-care delivery paradigm is required to provide accessible and inexpensive health services to the whole population [1]. Globally, most developed nations have generally adopted a three-tiered health-care service model and gatekeeping system in which patients visit primary care doctors or general practitioners for the treatment of common illnesses [2,3]. Then, the general practitioners send patients to hospitals and other medical service institutions for emergency and specialized treatment based on the patient’s condition. This is essential for the development of accessible, sustainable and equitable health systems; Britain’s National Health Service is one of the most cost-effective health services in the world because the general practitioner system acts as a gatekeeper [4].

China accounts for more than 20 percent of the world’s population, but only 2 percent of the world’s medical and health resources; the rapidly growing elderly population has caused health care resources to become scarce and difficult to access [5]. To address this, the Chinese government transformed the above system characteristics into the hierarchical diagnosis and treatment system (HDTS) in 2015, which guides patients to go firstly to primary health institutions, where those with severe diseases are referred to tertiary hospitals if necessary [6]. However, this ambitious national health care reform program in China has not solved the problem of “difficult and expensive medical treatment” [7,8,9]. There are data showing that China’s primary health care quality is still lacking. The visits of patients in primary medical institutions dropped from 4.34 billion to 4.12 billion from 2015 to 2020, whilst those of higher-level hospitals rose during the same period. Moreover, between 2015 and 2020, the average sickbed utilization rate of primary medical institutions was only 65%, while that of high-level hospitals reached as high as 95% [10]. The critical reason is that China’s HDTS does not require mandatory primary health care, and most patients choose to bypass primary medical institutions and enter general hospitals when they need treatment, resulting in a phenomenon that general hospitals are overcrowded while other medical institutions have insufficient patients [11,12,13,14]. The key to getting rid of the dilemma of graded diagnosis and treatment is for residents to choose grassroot medical treatment. Only by understanding the drivers and barriers on the intention of first visit in primary care institutions can more effective policies and interventions be developed to promote the first consultation at the grassroots level.

Patients’ health-seeking behavior and institution choice are part of a complex decision-making process. Previous literature suggests that they are not only affected by objective factors, such as demographic characteristics, economic environment, national policies, the distribution of medical institutions, and medical institutions’ conditions, but also by subjective factors, such as self-perceived health status, cognition of diseases and personal preference [15,16,17,18,19,20,21,22,23,24,25]. However, we currently know little about the role of information supply in the patients’ medical behavior or institution choice. In fact, the information supply may play an important role. As Arrow pointed, in the medical market, “information is a valuable commodity”, and changes in the structure of information supply and demand are likely to have an “inducing effect” on individual medical choices [26,27,28,29]. In particular, the rapid development of the Internet has caused the explosive growth of medical information and substantially increased the accessibility of medical knowledge. This makes the Internet one of the primary sources of medical information and expertise for residents prior to seeking medical treatment [30,31,32,33,34,35]. The Internet has exacerbated the spillover of medical information, and changes in the quantity, quality, and scope of information supply, which affect patients’ decision making [36,37].

Several research works involving Internet use suggest that the association between the Internet and medical decision making is still unclear. According to Lee et al., Internet health information has the potential to significantly influence the health attitudes and behaviors of a substantial part of the population, as well as the treatment of chronic illnesses [38]. A previous study, however, found that while the Internet might improve individuals’ health-related knowledge and attitudes, it seldom impacted their health-related actions [39]. Patients demonstrated interest in online comparative health care information in the study of Zwijnenberg et al., but the influence of the Internet on patients’ decision making remained restricted [40]. Consequently, it is still unclear whether searching online information through the Internet will affect patient’s health-seeking behavior and institution choice.

People browse health information through the Internet to make further medical decisions. On the one hand, the Internet provides diagnosis and treatment plans for almost all types of common diseases, including basic definition and symptoms, drug-use methods, and contraindications [41,42,43]. Studies have shown that an increasing number of people tend to use the Internet to obtain health care information [44,45,46], including older adults [47,48]. After the appearance of disease symptoms, patients use the Internet to search for the type of disease corresponding to the symptoms and determine the severity of the disease [49,50]. If the patient judges that the disease is common and not serious, he/she will purchase drugs and self-diagnose according to the online treatment plan. The Internet offers many advantages for patients in comparison with the offline world, such as convenience, time saving, and reduced limitations on space and time. The statistical report on Internet development has suggested that Internet penetration has continued to grow, and the popularity of the Internet has gradually spread to the elderly from the young [51]. Based on the above analysis, this paper assumes the following:

**Hypothesis** **1.**
*Internet use has a significant positive impact on self-treatment among older adults.*


On the other hand, due to the multi-source nature of Internet information, it may aggravate the incompleteness of the individual knowledge of the disease and make individuals more dependent on authoritative medical institutions [52]. In terms of rare diseases, it is difficult for different websites and platforms to provide patients with consistent information, and the uncertainty of diagnosis and treatment content may aggravate the risk perception of patients’ disease. Patients often have access to information that could psychologically prepare the latter but could also scare them. Online medical information focuses on universality and introductory content and does not list the probability of potential consequences. As a result, patients often overestimate the negative consequences of rare diseases and increase the expected utility loss of misdiagnosis [53]. As far as serious diseases are concerned, the difficulty and individualization make patients have to undergo equipment inspections and professional diagnosis [54]. Community health service institutions located in common and chronic diseases face cross-border competition from Internet medical information. Top-level hospitals are in a professional monopoly position because they are good at handling difficult and complicated diseases [55,56,57,58,59,60,61]. Based on the above analysis, this paper assumes the following:

**Hypothesis** **2.**
*When choosing medical facilities for common diseases, Internet use increases older adults’ preference for top-level hospitals.*


**Hypothesis** **3.**
*When choosing medical facilities for serious diseases, Internet use increases older adults’ preference for top-level hospitals.*


To test the above hypotheses, this paper applied several complementary methods, such as logistic regression, propensity score matching (PSM) and fixed-effect models (FE) based on a sample consisting of 9416 elderly participants from the China Health and Retirement Longitudinal Study (CHARLS), which is a multi-panel nationally representative household survey of the Chinese population aged 45 years and older conducted through multistage cluster sampling. Over the past 20 years, China has witnessed the rapid development and application of the Internet. By the end of 2020, the number of Internet users in China was 989 million, making China the largest Internet user in the world. Studying Internet medical information spillover in China is of particular interest, given that China is the world’s largest developing country and facing a serious ageing population issue, which will inevitably further lead to a significant increase in the demand for medical services.

Compared with the existing literature, the main contributions of this study are as follows. First, the present paper is (to the best of the author’s knowledge) the first attempt to comprehensively evaluate the impact of Internet use on the choice of health care. This study can contribute to a better understanding of the causes of medical decisions among older adults and provides a useful guide to strategy and policy formulation in the healthcare sector. Second, it provides a new analytical perspective for a hierarchical medical system, extending the research focus of grading diagnosis and treatment from institutional analysis and economic incentives to information induction. Third, this paper uses longitudinal panel data, mixed cross-sectional data, PSM and FE to solve the problem of selection bias and endogeneity to a large extent based on national survey data with a large sample size, and provides stronger statistical capabilities and more general conclusions.

## 2. Materials and Methods

### 2.1. Source of Data

The data used in this study were from the China Health and Retirement Longitudinal Study (CHARLS), which is a nationally representative survey of the population 45 years or above living in China, funded by Peking University (China), National Institute on Aging (China), and World Bank. Since 2011, CHARLS has conducted a survey every two years, sampling 28 (out of 31) provinces in China through multi-stage stratified probability proportionate to size sampling (PPS), which represents about 95% of China’s population. The database is public, and more detailed description of the sampling design and process can be obtained from its website (http://charls.pku.edu.cn, accessed on 1 February 2021). In each survey wave, about 17,000 people living in 10,000 households in 150 counties/districts and 450 villages/resident committees (or villages) were surveyed by using the face-to-face computer-assisted personal interview. Due to the long time span of the follow-up survey, CHARLS research is faced with some temporary or permanent exits, which are offset by the new respondents, that is, data imbalance. The survey aims to provide a database for population ageing academic research and public health policy analysis by tracking and collecting a wide range of information on demographic and socio-economic characteristics, family relations and dynamics, wealth, employment, education, health status and functioning, biomarkers, health care and insurance [62,63,64]. The Institutional Review Board of Peking University granted ethical consent (IRB00001052-11015). To research the association between the Internet use and health-seeking behavior and institution choice of the elderly population, we limited the samples to respondents who fell ill in the last month. This paper selects the mixed cross-sectional data of the three phases of 2011, 2013 and 2015 as the analysis object. Compared with the cross-sectional data, the mixed cross-sectional data can increase the sample size, expand the sample representativeness, and obtain more precise estimates and more effective statistics. By eliminating the missing values, the final sample contains 9253 individuals.

### 2.2. Variable Measurement

Self-treatment. It was measured by asking middle-aged and elderly respondents who fell ill in the last month whether they had self-medicated. If the respondent has self-medicated by buying over-the-counter Western medicine or prescription Western medicine, it is recorded as 1; otherwise, it is recorded as 0.

Health institution selection for common disease, HIS-CD. It was measured by asking the type of medical institution visited by the elderly respondents who fell ill in the last month. If the respondent attends a community health service center, township health center, health service station, village clinic or private clinic and other primary medical institutions, it is recorded as 0; if the respondent is in a general hospital, specialized hospital, and other non-primary medical institutions, it is recorded as 1.

Health institution selection for major disease, HIS-MD. It was measured by asking middle-aged and elderly respondents what type of medical institution they were hospitalized in most recently. If the respondent is hospitalized in primary medical institutions, such as community health service centers, township health centers, health service stations, village clinics or private clinics, it is recorded as 0; if the respondent is in general hospitals, specialized hospitals, and other hospitalization in non-primary medical institutions, it is recorded as 1.

Internet use, IU. Like previous studies that considered Internet use through smartphones or mobile phones [65,66,67,68], all respondents were asked whether they used a computer or mobile phone to surf the Internet. The value of the main treatment variable “Internet use” is 1 when respondents use the telephone or mobile phone to surf the Internet; otherwise it is 0. Because online browsing is a necessary condition for the overflow of Internet medical information, only individuals participating in online browsing activities have the opportunity to access Internet medical information. The process of online browsing greatly increases the possibility that individuals access medical information through the Internet. Therefore, the proxy variable is reasonable.

Control variable. Referring to Liu et al. and Zhang et al., this paper compiled a rich set of control variables, including age, gender, marital status, education level, urban or rural areas, physical health, IADL (Instrumental Activity of Daily Living), per capita household income, family members and children quantity [41,63]. Considering that the type of residence may influence the interviewees accessing medical information through the Internet and choosing the type of medical treatment, this paper controls the type of residence. Finally, this paper controls the regional effect to eliminate the influence of regional differences. Table 1 shows the descriptive statistics of these variables. In accordance with our expectation, compared with the non-Internet users, Internet users had a higher probability of self-treatment and select a top-level hospital for treatment of common diseases. However, there was no significant difference between Internet users and non-users in terms of medical options for treating severe cases. Of course, these descriptive statistics provide only instructive evidence. More rigorous analysis is needed to control for other confounding factors.

### 2.3. Analytic Strategy

Since the dependent variable in this study is a dichotomous variable, logistic regression analysis is used to establish the following measurement model:(1)$$\log\;(Y_{1}{}_{i})=\ln(p/(1-p))=\beta_{0}+\beta_{1}Internet_{i}+\sum_{j=1}^{N}\gamma_{j}Z_{i}+pr_{i}+\varepsilon_{i}$$
(2)$$\log\;(Y_{2}{}_{i})=\ln(p/(1-p))=\beta_{0}+\beta_{1}Internet_{i}+\sum_{j=1}^{N}\gamma_{j}Z_{i}+pr_{i}+\varepsilon_{i}$$
(3)$$\log\;(Y_{3}{}_{i})=\ln(p/(1-p))=\beta_{0}+\beta_{1}Internet_{i}+\sum_{j=1}^{N}\gamma_{j}Z_{i}+pr_{i}+\varepsilon_{i}$$

From Equation (1) to Equation (3), the subscript i is the *i*-th respondent; as the core explanatory variable, Internet is the usage of Internet. If the respondent often surfs the Internet, Internet = 1; if the respondent does not surf the Internet or does not surf the Internet frequently, Internet = 0. p is the event probability. Z is a series of personal characteristics and family characteristics control variables. pr is a province fixed effect, used to control regional differences. $\varepsilon_{i}$ represents the error term, independent and identically distributed. $\beta_{0}$ and $\beta_{1}$ are the parameters to be estimated; the dependent variable *Y*_1_ is the self-diagnosis and treatment choice of respondent *i*. *Y*_2_ is the health care choice of general disease. *Y*_3_ is the health care choice of serious illness.

In this study, whether respondents are online is not randomly assigned, but is the result of their conscious choice based on their own characteristics and resources. Therefore, there may be potential endogenous problems due to selection bias or missing variables. Therefore, this paper uses propensity score matching and a fixed-effects model to solve endogenous problems [69].

PSM is a common method for dealing with self-selection problems. The basic idea is to select a certain sample from the control group to match the sample in the intervention group according to the propensity score, and then estimate the treatment effect based on the resultant variable difference between the paired samples. The propensity score is the probability of the sample as the intervention group. It is estimated by the logit model after the observable variable X is given. The calculation equation is as follows:(4)$$P\left(X_{i}\right)=Pr\left(D_{i}=1|X_{i}\right)=\frac{e^{\beta x_{i}}}{1+e^{\beta x_{i}}}$$

In Equation (4), X is the control variable matrix, D is the indicator variable, the intervention group is 1, and the control group is 0. The common matching methods include kernel matching, nearest neighbor matching and radius matching. In order to ensure the robustness of the results, this paper adopts the nearest neighbor matching (1:1), nearest neighbor matching (1:5), kernel matching and radius matching, respectively. After matching, the influence degree of intervention on explained variables can be measured. The average treatment effect of intervention group is usually estimated, and its expression is as follows:(5)$$ATT=E(Y_{1}|D=1)-E(Y_{0}|D=1)=E(Y_{1}-Y_{0}|D=1)$$

In Equation (5), $Y_{1}$ represents the value of the explained variable when the intervention group sample receives the intervention, and $Y_{0}$ represents the value of the explained variable when the intervention group sample assumes no intervention.

This study may have potential endogenous problems, due to selection bias or missing variables. PSM can only solve the selection bias caused by observable variables, and potential missing variables may still bias the final estimation results. Therefore, this paper uses the 2011–2015 three-phase panel data and uses a fixed-effects model (FE) to control the unobserved personal characteristics and family characteristics that do not change over time. Since the data of the three periods are separated by only 2 years, some unobserved variables, such as health concepts and environmental preferences, are less likely to change during the three periods. The measurement equation of the fixed model (FE) of panel data is as follows:(6)$$Y_{it}=\beta_{0}+\beta_{1}Internet_{i}{}_{t}+\sum_{j=1}^{N}\gamma_{j}Z_{it}+province_{i}+{\mathrm{year}}_{t}+\delta_{i}+\varepsilon_{i}{}_{t}$$

In Equation (6), the subscript i is the i-th respondent; t is the year; as the core explanatory variable, Internet is the usage of Internet; Z is a series of personal characteristics and family characteristics control variables; province is the province fixed effect; year is the year fixed effect; $\varepsilon_{i}$ represents the error term, independent and identically distributed; $\beta_{0}$ and $\beta_{1}$ are the parameters to be estimated; the dependent variable $Y_{1}$ is the self-diagnosis and treatment choice of respondent *i*. $Y_{2}$ is the choice of respondent *i*‘s general disease clinic. $Y_{3}$ is the choice of respondent i’s serious illness clinic.

## 3. Results

### 3.1. Benchmark Regression Results

Before conducting an empirical analysis, we ran a multicollinearity test. The maximum value was 4.13, which was far below the experience VIF value of 10. Therefore, we can confirm that multiple collinearity did not have much of an effect on the regression analysis. Table 2 presents the results from the estimation of specification (1). In this paper, according to Equations (1)–(3), logit estimates are made on the effect of medical options on using the Internet, and the results are shown in Table 2. The results in column (1) show that Internet use has a significant positive impact on the self-treatment of common diseases (β = 0.05, *p* < 0.05). Specifically, using the Internet can increase the probability of the elderly self-treatment by 5%, so Hypothesis 1 is supported. The results in column (2) show that Internet use has a significant positive impact on the elderly choosing top-level hospitals for the treatment of common diseases (β = 0.06, *p* < 0.01), so Hypothesis 2 is supported. However, the results in column (3) show that Internet use has no significant impact on the elderly choosing top-level hospitals for the treatment of major disease (β = 0.03, *p* > 0.1). Hypothesis 3 is not supported.

### 3.2. PSM Results

PSM is a common method for dealing with self-selection problems. The basic idea is to select a certain sample from the control group to match the sample in the intervention group according to the propensity score, and then estimate the treatment effect based on the resultant variable difference between the paired samples [70,71]. The logit probability model to estimate the conditional probability of respondents is adopted. (The estimation results of the logit probability model show that the probability of Internet use is significantly correlated with age, gender, urban–rural differences, education level, family income per capita, IADL, number of children, and housing type. Due to space limitations, detailed estimation results are not reported in this paper.) To ensure that different matching methods do not interfere with the estimation results, the nearest neighbor matching method (1:1), nearest neighbor matching method (1:5), kernel matching method and radius matching method are adopted, respectively. Only the samples with the most similar propensity scores of the intervention group are kept in the control group, while those that do not match are deleted. Next, the matching quality needs to be tested. This paper first compares the propensity score overlap between the intervention group and the control group before matching, as shown in Figure 1a. In general, the propensity score distribution of the intervention group is significantly different from that of the control group. Figure 1b shows the distribution of propensity scores after matching. (This paper adopts a variety of matching methods for analysis, and all pass the matching quality test. Due to space limitations, only the nearest neighbor matching method (1:5) test results are reported here.) It can be intuitively judged that propensity score matching obviously corrects the score deviation between the two groups.

This paper firstly estimates the average treatment effect before matching, and the results are shown in Table 3. The average treatment effect before matching is significantly higher than that after matching, which means that if the selection bias is not considered, the influence of the Internet on the choice of medical treatment will be overestimated. There are four different matching methods adopted to calculate the average treatment effect. The estimation results of different matching methods are basically the same, indicating that this study is not sensitive to matching methods and has good robustness. This paper takes the average of the average treatment effects of different estimation methods for subsequent analysis.

### 3.3. Heterogeneity Analysis

The analysis in this paper shows that Internet use has significantly increased the probability of treatment for middle-aged and elderly people, but at the same time, it has increased the flow of middle-aged and elderly people to high-level medical institutions for common diseases. However, the influence of the Internet on medical choice may vary from group to group. For this reason, this paper conducts an in-depth analysis of the influence of different groups.

First, this paper investigates whether the effects of gender differences are heterogeneous. The results are shown in Table 4. The usage of Internet has a significant positive impact on the self-treatment of common diseases in middle-aged and elderly men, but the impact on women is not significant. It may be caused by gender differences in sociology or psychology; the usage of the Internet has a significant positive impact on men and women choosing high-level medical institutions for general disease treatment.

Second, this paper investigates whether the impact of urban–rural differences is heterogeneous, and the results are shown in Table 4. The usage of the Internet has a significant positive impact on the self-treatment of middle-aged and elderly people in rural areas, but it has no significant impact on middle-aged and old people in urban areas. This is because middle-aged and old people in urban areas generally conduct self-treatment when facing common diseases. (According to the regression results in Table 2, the coefficient of impact of living in a city on self-treatment of common diseases is 0.86 (significant at the 1% level).) Therefore, the further impact of the Internet is limited. The usage of the Internet has a significant positive impact on rural and urban middle-aged and elderly people choosing high-level medical institutions for general disease treatment, while the usage of the Internet also has a significant positive impact on rural middle-aged and elderly people.

Finally, we studied whether the influence of different economic groups is heterogeneous. Based on the difference in household income per capita, we divided the respondents into high-income, middle-income and low-income groups. (The income group is divided according to the 2015 poverty standard issued by the National Bureau of Statistics of China. Respondents whose per capita household income is less than CNY 2700 are considered low income, and those whose household per capita income is between CNY 2700 and CNY 20,000 are considered middle income. Those with a per capita income of more than CNY 20,000 are considered high income). The results are shown in Table 4. The usage of the Internet only affects the choice of medical treatment for middle-income groups but has little effect on low-income and high-income groups. The explanation for this is that low-income groups cannot afford higher medical expenditures. Therefore, after illness, low-income groups tend to use cheap traditional Chinese medicine. Even if they have to go to the hospital, they choose lower-cost primary medical institutions. Therefore, Internet use has a limited impact on them. The high-income groups pay more attention to their own health conditions. They generally conduct self-treatment after illness and choose higher-level hospitals for treatment. Therefore, Internet use will no longer affect the medical choices of high-income groups.

### 3.4. Robustness Test

In view of the fact that PSM can only solve the problem of selection bias caused by observable variables, and the omitted variables may still interfere with the final estimation results, this paper further adopts the fixed-effects model (FE) to solve the endogenous problem. The advantage of the FE model is that it can eliminate the heterogeneity that does not change over time. However, because it only estimates the “in-group variation” of the variable, it may cause a large variance bias and reduce the estimation accuracy. For this reason, this paper also carried out random effects model estimation, and the results are shown in Table 5. Before conducting an empirical analysis, we ran a multicollinearity test. The maximum value was 3.31, which was far below the experience VIF value of 10. The estimated results of the FE model and the random-effects model on the impact of Internet on self-treatment, general disease treatment institution selection, and severe disease treatment institution selection are very similar to the results of PSM in Table 3, which shows that the results of this paper are robust.

## 4. Discussion

This paper investigated the impact of Internet use on medical decisions medical decisions among Chinese older adults through several complementary methods, such as logistic regression, propensity score matching (PSM) and fixed-effect models (FE) based on the 2011, 2013, and 2015 China Health and Retirement Longitudinal Study (CHARLS). The results showed that the Internet had a certain effect on older adults’ health seeking behavior and institution choice. First, the elderly with Internet behavior are more inclined to self-cure when suffering from common diseases, especially for rural residents and middle-income groups, indicating that self-diagnosis and therapy can partially replace hospital care. Second, in terms of medical institution choices, those who use Internet are more inclined to choose top-level hospitals than community health service institution to treat common diseases. The study contains theoretical and practical consequences for how to govern Internet health care and direct people to medical institutions, as well as a reference to Internet medical treatment promotion and implementation.

Chinese older adults who use Internet are more inclined to self-treatment than visiting hospitals, which is consistent with some research descriptions. Yang et al. pointed out that online medical platforms have become the “entrance” for many patients to see a doctor, which, to a certain extent, diverts the flow of patients with common diseases to high-level hospitals [72]. As the popularity of the Internet has grown, surfing and selecting health information has become a standard procedure before deciding whether or not to visit a hospital [73]. The popularity of the Internet and mobile Internet has broken the medical information barriers, and the public can obtain diagnosis and treatment measures for common diseases from the Internet at low cost and conveniently. The emergence of online appointment registration services, online health care and monitoring, telemedicine, online diagnostic and treatment services, and medical supplies businesses related to medical services, medicines, and online consultations enable people to enjoy online medical services more quickly, efficiently, and at low cost [74]. Overall, the Internet has the potential to minimize barriers to common illness knowledge, to some extent lessen information asymmetry between patients and physicians, and increase individuals’ awareness and access to fundamental health knowledge, lowering the likelihood of utilizing medical services. Due to the current shortage of medical resources and the public’s thirst for medical resources, the integration of medical resources and the Internet is an important way to improve China’s lack of medical resources.

In contrast, this study discovered that the Internet may increase the likelihood of seeking medical treatment from the top-level hospitals. The multi-source and uncertainty of medical information acquisition has exacerbated the inconsistency and incompleteness of an individual’s perception of disease. Due to the limitation of professional knowledge, it is difficult for patients to identify the relevant information, and they are prone to be misled by the wrong medical information, which leads to health anxiety; for instance, physical symptoms are misinterpreted as signals of dangerous diseases, and there is a continual worry of being sick [75]. Ogasawara found that many websites offer harmful information about cancer, and the proportion of these websites is far higher than that of sites that offer reliable information about cancer treatment [76] which brings “noise” and intensifies the “increasing tendency” of regular medical institutions. If the development of Internet medical care is allowed to develop savagely and irregularly, it may further aggravate the medical burden of hospitals. The competent government departments should strengthen the supervision and guidance on the quality of medical information on the Internet to ensure the authority of medical information.

This study also has some limitations. First, although this paper has largely solved the endogenous problem caused by selection bias and missing variables that do not change over time through PSM and FE, but due to data limitations, no suitable instrumental variables were found to further ensure the rigor of the results. Second, this is a survey on the middle-aged and elderly people in China. Because the health of the middle-aged and old people is quite different from that of other groups, all the conclusions of this study cannot be generalized to the general group. With the further development of online medical care, research on the mechanism of the influence of patients’ medical behavior from the perspective of the Internet will be the direction of future research.

## 5. Conclusions

With the rapid development of a new generation of information technology, the dissemination and utilization of medical service information has accelerated, and the functions of online medical services have also continuously expanded. The Internet has broken down the barriers to the knowledge of common diseases, shortened the gaps in health information accessibility, and has produced a slight substitution effect of self-diagnosis and treatment on hospital care. However, the knowledge monopoly of difficult and complicated diseases cannot be eliminated, and at the same time, the increase in inconsistent, incomplete, and commercialized medical information has also brought noise to decision making, blurring the residents’ cognitive boundary of common diseases and severe diseases. Consequently, the rising tendency of visiting high-level medical institutions may be exacerbated, which will be unable to guide patients to hierarchical diagnosis and treatment. The government should issue relevant policies to regulate the development of Internet medical care and guide patients to choose reasonable medical institutions based on their own conditions so as to achieve the purpose of hierarchical diagnosis and treatment, save costs, and greatly improve service efficiency and service quality.

## Figures and Tables

**Figure 1 healthcare-10-00060-f001:**
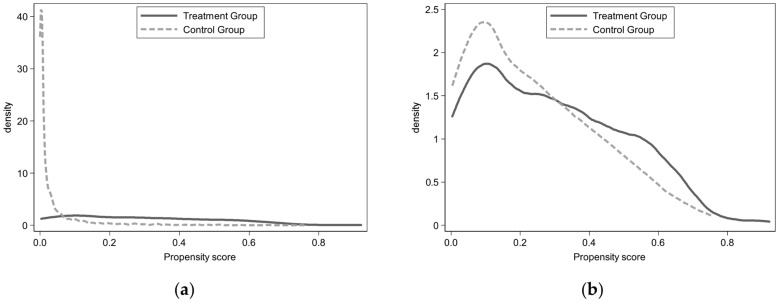
Probability score distribution. (**a**) Before matching; (**b**) after matching.

**Table 1 healthcare-10-00060-t001:** Descriptive statistics.

Variables	Definition	ALL	Internet Users	Non-Internet Users	Mean Difference
		Mean (S.D.)	Mean (S.D.)	Mean (S.D.)	
Self-treatment	0 = no self-treatment, 1 = self-treatment	0.468 (0.499)	0.490 (0.495)	0.465 (0.499)	0.025 ***
HIS-CD	0 = primary medical institution, 1 = great public hospitals	0.366 (0.482)	0.473 (0.470)	0.354 (0.478)	0.119 ***
HIS-MD	0 = primary medical institution, 1 = great public hospitals	0.805 (0.397)	0.824 (0.267)	0.801 (0.400)	0.023
Age	ranging from 45 to 101 years	59.62(9.734)	53.52(7.272)	59.86 (9.740)	−6.34 ***
Gender	0 = male, 1 = female	0.516 (0.500)	0.408 (0.492)	0.521 (0.500)	−1.113 ***
Urban vs. rural residence	0 = rural residents, 1 = urban residents	0.212 (0.409)	0.708 (0.455)	0.192 (0.394)	0.516 ***
Marriage status	0 = having no partner, 1 = having a partner	0.871 (0.336)	0.926 (0.262)	0.869 (0.338)	0.057
Education	1 = uneducated, 2 = literate, 3 = primary education, 4 = secondary education, 5 = tertiary education	2.846 (1.362)	4.543 (0.698)	2.778 (1.338)	1.765 ***
Insured	0 = having no insurance, 1 = having insurance	0.937 (0.243)	0.958 (0.201)	0.936 (0.245)	0.022
Self-rated health	self-rated health: 0 = poor, 1 = fair, 2 = good, 3 = very good, 4 = excellent	2.165 (0.581)	1.912 (0.500)	2.175 (0.582)	−0.263 ***
IADL	Instrumental Activity of Daily Living, 0–28	10.18 (4.056)	7.663 (2.677)	10.28 (4.069)	−2.6171 ***
Per capital income	log form of annual household per capital income (CNY)	8.106 (2.612)	9.408 (2.622)	8.054 (2.598)	1.354 ***
Household size	the total number of family members	3.258 (3.583)	3.247 (1.515)	3.259 (3.641)	−0.012
Number of children	the total number of children in the legal sense	2.792 (1.446)	2.656 (0.870)	2.837 (1.446)	−0.181 ***
Housing type	0 = wood, bamboo, grass, sheet iron, cave dwelling, adobe, 1 = bricks and wood, mixed structure, 3 = concrete and steel	1.355 (0.701)	1.564 (0.726)	1.347 (0.699)	0.217 ***
Observations		9416	1884	7528	

Note. * *p* < 0.1, ** *p* < 0.05, *** *p* < 0.01.

**Table 2 healthcare-10-00060-t002:** Effect of Internet use on medical decisions.

Variables	(1)	(2)	(3)
	Self-Treatment	HIS-CD	HIS-MD
Internet use	0.050 **	0.060 ***	0.030
	(0.047)	(0.001)	(0.516)
Age	−0.001	−0.010	−0.010 *
	(0.588)	(0.221)	(0.066)
Gender	0.030	−0.200 ***	−0.080
	(0.734)	(0.004)	(0.498)
Urban vs. rural residence	0.360 ***	0.860 ***	1.290 ***
	(0.003)	(0.000)	(0.000)
Marry	0.030	−0.080	0.080
	(0.760)	(0.426)	(0.587)
Uneducated	0.000	0.000	0.000
	(.)	(.)	(.)
Literate	0.330 **	0.240 **	0.180
	(0.011)	(0.031)	(0.274)
Primary education	0.220 **	0.220 **	0.320 **
	(0.050)	(0.027)	(0.033)
Secondary education	0.270 **	0.460 ***	0.450 **
	(0.040)	(0.000)	(0.015)
Tertiary education	0.120	0.650 ***	0.830 ***
	(0.493)	(0.000)	(0.001)
Insured	0.420 ***	−0.100	−0.290
	(0.004)	(0.507)	(0.327)
Per capital income	0.040 ***	0.040 ***	0.040 **
	(0.002)	(0.004)	(0.030)
Self-rated health	0.200 ***	0.000	−0.150
	(0.006)	(0.989)	(0.114)
IADL	0.008	0.040 ***	0.050 ***
	(0.011)	(0.000)	(0.000)
Household Size	−0.020	0.000	0.010
	(0.359)	(0.818)	(0.788)
Number of children	0.020	−0.060 **	0.080 *
	(0.583)	(0.035)	(0.068)
Humble and old house	0.000	0.000	0.000
	(.)	(.)	(.)
Brick and concrete house	0.040	−0.210 **	−0.380 **
	(0.747)	(0.042)	(0.044)
Concrete and steel house	−0.070	0.080	−0.180
	(0.548)	(0.424)	(0.311)
Constant	−2.360 **	−2.680 ***	−0.160
	(0.022)	(0.635)	(0.861)
Province fixed effects	YES	YES	YES
Year fixed effects	YES	YES	YES
Observations	3314	4938	2444
Wald	136.490	601.820	183.370
R-squared	0.1381	0.1199	0.110

Note. * *p* < 0.1, ** *p* < 0.05, *** *p* < 0.01, robust standard errors in parentheses. HIS-CD = health institution selection-common disease; HIS-MD = health institution selection-major disease.

**Table 3 healthcare-10-00060-t003:** The average treatment effect of Internet on medical decisions.

Matching Method	(1)	(2)	(3)
	Self-Treatment	HIS-CD	HIS-MD
Before the match ATT	0.111 ***	0.294 ***	0.126 **
	(0.041)	(0.032)	(0.044)
Nearest neighbor matching (1:1) ATT	0.099 **	0.116 **	0.063
	(0.064)	(0.053)	(0.052)
Nearest neighbor matching (1;5) ATT	0.093 **	0.117 **	0.017
	(0.049)	(0.041)	(0.037)
Radius matching method ATT	0.081 **	0.116 ***	0.027
	(0.046)	(0.038)	(0.034)
Kernel matching ATT	0.079 **	0.115 **	0.026
	(0.046)	(0.038)	(0.034)
ATT Average	0.088 **	0.116 **	0.033

Note. * *p* < 0.1, ** *p* < 0.05, *** *p* < 0.01, AI robust standard errors in brackets. ATT = average treatment effect. The average treatment effect of this paper: controlled age, gender, marital status, education level, urban-rural differences, family income per capita, health status, IADL, number of family members, number of children, housing type, and controlled urban fixed effects and time fixed effects.

**Table 4 healthcare-10-00060-t004:** Heterogeneity analysis of the influence of Internet on medical decisions.

Variables		(1)	(2)	(3)
		Self-Treatment	HIS-CD	HIS-MD
Gender	Male	0.142 **	0.131 **	0.018
		(0.067)	(0.060)	(0.054)
	Female	0.028	0.112 *	0.05
		(0.080)	(0.062)	(0.056)
Urban vs. rural residence	Rural	0.148 **	0.118 *	0.161 **
		(0.077)	(0.070)	(0.071)
	Urban	0.020	0.089 *	−0.003
		(0.060)	(0.048)	(−0.09)
Family economic status	Low income	−0.016	−0.130	0.001
		(0.164)	(0.164)	(0.19)
	Middle income	0.174 ***	0.192 ***	0.062
		(0.060)	(0.069)	(0.054)
	High income	0.088	0.072	−0.02
		(0.084)	(0.055)	(0.044)

Note. * *p* < 0.1, ** *p* < 0.05, *** *p* < 0.01, AI robust standard errors in brackets. the nearest neighbor matching (1:5) was used for estimation. The average treatment effect also controls age, gender, marital status, education level, urban–rural differences, per capita household income, health status, IADL, number of family members, number of children, and housing type; the fixed effects of city and time, limited to length, are no longer reported.

**Table 5 healthcare-10-00060-t005:** Estimates of fixed effects and random effects of the influence of Internet on medical decisions.

Variable	(1)	(2)	(3)
	Self-Treatment	HIS-CD	HIS-MD
Panel A: Fixed-effects model			
Internet use	0.070 **	0.155 **	0.02
	(0.071)	(0.025)	(0.040)
Control variable	yes	yes	yes
Personal fixed effect	yes	yes	yes
Year fixed effect	yes	yes	yes
N	3987	7810	2945
F	0.27	4.28	3.90
Within R-sq	0.023	0.0267	0.059
Panel B: Random effects model			
Internet use	0.078 ***	0.092 **	−0.013
	(0.019)	(0.010)	(0.041)
Control variable	yes	yes	yes
Urban fixed effect	yes	yes	yes
Year control variable	yes	yes	yes
N	3987	7810	2945
Within R-sq	0.179	0.233	0.152
Between R-sq	0.418	0.384	0.346
Overall R-sq	0.021	0.1310	0.093

Note. * *p* < 0.1, ** *p* < 0.05, *** *p* < 0.01, the parentheses are the Driscoll–Kraay error. The fixed-effects model has two-way control of time effect and individual effect. It also controls age, gender, marital status, education level, urban–rural differences, family income per capita, health status, IADL, number of family members, number of children, and housing type; the random effects model also controls these variables and controls the province fixed effects. Due to space limitations, this paper only reports the estimated results of the key explanatory variables Internet use.

## Data Availability

Not applicable.

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
