# Peer review of "Does Internet Use Affect Medical Decisions among Older Adults in China? Evidence from CHARLS"

_healthcare, 2021, doi:10.3390/healthcare10010060_

Round 1

Reviewer 1 Report

Reviewer Comments to Author:

The study is quite interesting, and the outcomes of this paper are a valuable addition to the literature (after major changes/revision). This study aims to explore the influence of Internet use on the health care choice of Chinese elder groups. Overall, this article is not written in a clear way, but it should not be granted publication in Healthcare at the present form of the manuscript. There are minor problems with English, i.e., Punctuations, grammar, and propositions. The paper needs copy editing. In addition, critical literature, hypotheses development and conclusion section, are major and significant concerns of this article and authors used the personal pronoun ‘we’ or ‘our’ throughout the paper and should replace with the word “this study or this research” See comments below for details.

The below are major concerns and area of improvement with suggestions.

  1. Originality: The paper aims to explore the influence of Internet use on the health care choice of Chinese elder groups. There is a need to revise the paper title, and the current title does not portray the actual meaning of paper.

  1. Abstract: The abstract is not well written. There is a need to revise with explicit contents of the abstract, i.e., the main issue, sampling, a statistical tool, methods, results, and implication. Author’s should provide precisely and focused abstract.

  • What is the practical and theoretical contribution of this article into literature?
  • The sampling criteria, population, and unit of analysis are missing. The author should highlight the sampling criteria for more clarity to readers.
  • As a suggestion for improvement, the author should not use the same Keywords as like Paper Title. It is encouraged to use different keywords which are not in the Paper title. It will enhance paper searchability after the publication.

  1. Introduction: The introduction section is not well written. There are ambiguous statements and no clarity in the introduction section.

  • The introduction section is not started with a broader area and issue or in a global context. There is no synthesis in writing an introduction section.
  • There is less debate on the targeted country problem, i.e., health sector problems. It will more value addition in the paper if the author explains with some statistics figures and recent issues.
  • In the Introduction section, the brief discussions of methods, tools, sampling, and findings are missing.
  • An important question to answer is, “Why should Healthcare readers be interested in the results of this paper, which scrutinized only one country data?” The reason given is not supportive. Are the findings generalizable to other developing countries like India, Pakistan, Malaysia, and Indonesia? The author(s) needs to improve underwritten motivation.

  1. Relationship to Literature: The paper did not incorporate major literature on internet usage in the health sector, and the paper does not sufficiently cover recent research in the area. Helpful in this regard would be to include relevant research recently covered in top journals of similar scope. Further, work needs to be done to clearly support the findings based on the current literature, as a recent theory in the area is directly counter to what was found.
  • Author(s) should use underpinning theory to justify this research. However, the major concern of this paper, the author(s) should highlight how this research is contributing to theory or contribution of the theory. Authors did not discuss much with the use of developed theories in this area.
  • There is a need to add more critical recent literature and on the basis of theoretical argumentation.
  • There should be a separate section for Literature review and must discuss all studied variables.
  • The hypotheses development is not written; author(s) should cite previous studies relevant to proposed hypotheses, i.e., international and local perspectives studies in the light of underpinning theory.

  1. Methodology: Author did not develop its argument from appropriate theory. However, they explored models previously studied in the same area. However, the data is focused on elderly people. Therefore, the article represents secondary data from the content technique as the sample and methodology and relevant to present the context of the proposed theme. However, the empirical sample was fair enough to illustrate significant empirical results of this paper, and the methods employed are not appropriate. In addition, there are major concerns with sampling design and data nature.

  • Author(s) obtained corporate governance information by the secondary source from regulatory authorities from 2011 to 2015. Author(s) did not highlight the following information.
  • What is the population of banks (Sampling frame, total population, )?
  • Why specifically were this period of 2011 – 2015 selected for this study?
  • Why have China been chosen?
  • What are the criteria for the selection of firms for data collection? (Sampling techniques, i.e., random, cluster, or judgmental sampling)?
  • Regarding the methodology, more details and justification of why they did not employ panel data analysis?
  • The author(s) did not define the data collection and sampling clearly.

  1. Results: The analysis is clearly provided to tie up with the findings.

  • The author should provide a complete statistical analysis, i.e., skewness, and kurtosis in descriptive analysis, and correlation analysis.
  • Results are poorly reported in the result section, and authors should discuss results with the help of statistical figures.
  • There is a need for improvement in reporting results such as the author(s) should report [Beta value OR standard error (S.E) with significance level OR t-value; i.e., (β= xxx. P<0.01) OR (S. E= xxx, t > xxx). However, it could be more effective if the author(s) presents significant results with bold and asterisk (*).
  • Why there is no diagnostic checks for panel data, i.e., VIF, auto correlation etc

  1. Discussion and findings: As results are clearly provided. However, there is no solid discussion on results.

  • Author(s) should discuss the limitations of this study and future research direction in a constructive way. Hence, authors should write in prices and in a constructive way under a subsection of discussion and results.
  • Author(s) did not discuss the theoretical and practical contribution of this study. Author(s) should discuss the theoretical and practical contribution of this study in the separate subsection under discussion for more clarity.

8- Conclusion: Author should provide concluding statements rather that repetitive statements in the conclusion portion. Abstract and conclusion sections look similar, please revised conclusion section.

  1. Citation and End References
  • The author did not cite the latest literature relevant to the target issue in this paper. Only a few articles are cited, which are published in the last five years. However, I strongly recommend more latest studies to cite and strengthen this paper.

  1. Quality of Communication: The paper needs further proofreading. I suggest that more careful investigation of prior literature can make this paper distinguishable. Linking this article with prior studies does not seem to be sufficient, which weakens the justification of incremental contributions.

Suggested Revisions

  • Abstract revision (complete / Major concern)
  • Sampling and Data (Need to explain in detail / Major concern)
  • Introduction Section (Please re-write holistically)
  • Need to add literature section
  • There is a need to discuss the theoretical and practical contribution of this study in separate subsections.
  • Conclusion section need to add and revise

I hope that the comments provided can help in this regard.

Author Response

  1. Originality:The paper aims to explore the influence of Internet use on the health care choice of Chinese elder groups. There is a need to revise the paper title, and the current title does not portray the actual meaning of paper.

Author response: We deeply appreciate your thoughtful comments. According to your suggestion, we have revised the title of the paper to “Does Internet use affect medical decisions among older adults in China? evidence from CHARLS” which portray the actual meaning of the paper more clearly.

  1. Abstract:The abstract is not well written. There is a need to revise with explicit contents of the abstract, i.e., the main issue, sampling, a statistical tool, methods, results, and implication. Author’s should provide precisely and focused abstract. What is the practical and theoretical contribution of this article into literature?The sampling criteria, population, and unit of analysis are missing. The author should highlight the sampling criteria for more clarity to readers. As a suggestion for improvement, the author should not use the same Keywords as like Paper Title. It is encouraged to use different keywords which are not in the Paper title. It will enhance paper searchability after the publication.

Author response: Thank you for your preciseness. According to your suggestion, we have revised the abstract. The modified abstract includes four parts: background, method, results and conclusion, which provide precisely contents of the abstract. The rewritten summary is as follows: “Background: The rapid growth of the elderly population poses a huge challenge for people to access medical services. The key to get rid of the dilemma is for patients to go firstly to primary medical institutions. Existing studies have identified numerous factors that can affect patients' health institution choice. However, we currently know little about the role of Internet use in the patients' medical decisions. The objective of this study is to explore health seeking behavior and institution choice under the background of the Internet era from the perspective of older adults, and to analyze whether the Internet could guide patients to the appropriate medical institution, so as to accomplish hierarchical treatment. Methods: The dataset comprises 9416 people aged 45 or above from the China Health and Retirement Longitudinal Survey (CHARLS) which through multistage cluster sampling conducted in 2011, 2013, and 2015. Logistic regression, PSM, and FE model are used to estimate the the influence of Internet use on the health care decision-making behavior. Results: Internet use has a significant positive impact on the self-treatment of common diseases (β=0.05, p<0.05). In terms of medical institution choices, the who use Internet were more inclined to choose top-level hospitals than community health service institution to treat common diseases (β=0.06, p<0.01). Conclusions: The Internet has lowered the obstacles to learning about common ailments, resulting in a substitution impact of self-treatment for hospital care. But Internet may aggravate older adults' perception of the risk of disease, which exacerbate the tendency of going to higher-level medical institutions for medical treatment. The finding of the study is useful for further rational planning and utilization of the internet, in order to guide patients to appropriate medical institution, which helps to improve the efficiency of the overall medical and health services.

According to your suggestion, we make it clear that the contribution of this article into literature in abstract. We supplemented the following content: “The finding of the study is useful for further rational planning and utilization of the internet, in order to guide patients to appropriate medical institution, which helps to improve the efficiency of the overall medical and health services.

According to your suggestion, we have supplemented the sampling criteria, population, and unit of analysis. We supplemented the following content: “The dataset comprises 9416 people aged 45 or above from the China Health and Retirement Longitudinal Survey (CHARLS) which through multistage cluster sampling conducted in 2011, 2013, and 2015.

Thank you for your carefulness. We have revised the keywords. The modified keywords include gate keeping system, self-treatment, primary health institutions, CHARLS, PSM, which are not in the Paper title.

  1. Introduction: The introduction section is not well written. There are ambiguous statements and no clarity in the introduction section. The introduction section is not started with a broader area and issue or in a global context.There is no synthesis in writing an introduction section. There is less debate on the targeted country problem, i.e., health sector problems. It will more value addition in the paper if the author explains with some statistics figures and recent issues. In the Introduction section, the brief discussions of methods, tools, sampling, and findings are missing. An important question to answer is, “Why should Healthcare readers be interested in the results of this paper, which scrutinized only one country data?” The reason given is not supportive. Are the findings generalizable to other developing countries like India, Pakistan, Malaysia, and Indonesia? The author(s) needs to improve underwritten motivation.

Author response: We deeply appreciate your thoughtful comments. According to your suggestion, we have rewritten the introduction section.

The new introduction starts from the global context, revealing that the world is facing the shortage of medical resources caused by aging, and there is an urgent need for an effective health care service delivery model to provide accessible and affordable health care services to all human beings. From a global perspective, most developed countries have widely implemented a three-level health care service model and gate keeping system. Just as Britain's National Health Service is one of the most cost-effective health services in the world, because of the general practitioner system acts as a gatekeeper.

Based on the national conditions, the Chinese government has transformed such system characteristics into the hierarchical diagnosis and treatment system (HDTS).  However, this ambitious national health care reform program in China has not solved the problem of “difficult and expensive medical treatment”. China's HDTS does not require mandatory primary health care, and most patients choose to bypass primary medical institutions and enter general hospitals when they need treatment, resulting in a phenomenon that general hospitals are overcrowded and other medical institutions with insufficient patients. The key to get rid of the dilemma of graded diagnosis and treatment is for residents to choose grass-root medical treatment. Only by understanding the drivers and barriers on the intention of first visit in primary care institutions can more effective policies and interventions be developed to promote the first consultation at the grassroots level. 

Although existing studies have identified numerous factors that can affect patients' health institution choice. However, we currently know little about the role of information supply in the patients' medical behavior or institution choice. The internet has caused the explosive growth of medical information and has greatly improved the availability of medical knowledge. This makes the internet one of the main ways for residents to obtain medical information and knowledge before seeking medical treatment. However, little has been researched on how the internet affects medical decisions. The objective of this study is to explore health seeking behavior and institution choice under the background of the Internet era from the perspective of older adults, and to analyze whether the Internet could guide patients to the appropriate medical institution, so as to accomplish hierarchical treatment.

According to your suggestion, we explain with some statistics figures and recent issues. We supplemented the following content: “There are data showing that China’s good primary health care quality is still lacking. The visits of patients in primary medical institutions dropped from 4.34 billion to 4.12 billion from 2015 to 2020, whilst that of higher-level hospitals have been rising during the same period. Moreover, between 2015 and 2020, the average sickbed utilization rate of primary medical institutions was only 65%, while that of high-level hospitals reached as high as 95%.

According to your suggestion, We have discussed the of methods, tools, sampling. We supplemented the following content: “This paper applied several complementary method such ass logistic regression, propensity score matching (PSM) and fixed-effect models (FE) base on a sample consisting of 9416 elderlies from the China Health and Retirement Longitudinal Study (CHARLS) which a multi-panel nationally representative household survey of the Chinese population aged 45 years and older through multistage cluster sampling.

According to your suggestion, We have answered the question “Why should Healthcare readers be interested in the results of this paper, which scrutinized only one country data?”. We supplemented the following content: “ Over the past 20 years, China has witnessed the rapid development and application of the Internet. By the end of 2020, the number of Internet users in China was 989 million, making China the largest Internet user in the world. Studying internet medical information spillover in China is of particular interest given that China is the world's largest developing country and facing serious aging population which will inevitably further lead to a significant increase in the demand for medical services.” 

We further illustrate the contribution of this study compared with previous studies. We supplemented the following content: “First, the present paper is (to the best of the author’s knowledge) the first attempt to comprehensively evaluate the impact of Internet use on the choice of health care. This study can contribute to a better understanding of the causes of medical decisions among older adults and provides a useful guide to strategy and policy formulation in the healthcare sector. Second, it provides a new analytical perspective for hierarchical medical system, extending the research focus of grading diagnosis and treatment from institutional analysis, economic incentives to information induction. Third, this paper uses longitudinal panel data and mixed cross-sectional data and PSM and FE to solve the problem of selection bias and endogenous to a large extent and based on national survey data with a large sample size and provides stronger statistical capabilities and more general conclusions.”

  1. Relationship to Literature: The paper did not incorporate major literature on internet usage in the health sector, and the paper does not sufficiently cover recent research in the area. Helpful in this regard would be to include relevant research recently covered in top journals of similar scope. Further, work needs to be done to clearly support the findings based on the current literature, as a recent theory in the area is directly counter to what was found.Author(s) should use underpinning theory to justify this research. However, the major concern of this paper, the author(s) should highlight how this research is contributing to theory or contribution of the theory. Authors did not discuss much with the use of developed theories in this area. There is a need to add more critical recent literature and on the basis of theoretical argumentation. There should be a separate section for Literature review and must discuss all studied variables. The hypotheses development is not written; author(s) should cite previous studies relevant to proposed hypotheses, i.e., international and local perspectives studies in the light of underpinning theory.

Author response: We deeply appreciate your thoughtful comments. Considering that none of the articles published in this journal contains an independent literature review section, we integrate the literature review section into the introduction. 

According to your suggestion, we have supplemented relevant literature on internet usage in the health sector. We supplemented the following content: “Several research involving internet use suggest that the association between the internet and medical decision making is still unclear. According to Lee et al, internet health information has the potential to significantly influence the health attitudes and behaviors of a substantial part of the population, as well as the treatment of chronic illnesses [38]. An previous study, however, found that while the internet might improve individuals' health-related knowledge and attitudes, it seldom impacted their health-related actions [39]. Patients demonstrated interest in online comparative health care information in Zwijnenberg et al's study, but the influence of the internet on patients' decision-making remained restricted [40]. Consequently, it is still unclear whether searching online information through the internet will affect patient’s health seeking behavior and institution choice.

Thank you for your preciseness. Since few studies have focused on the impact of Internet use on medical decision-making, we did not find a mature theory in this field applicable to this study. This study is an exploratory study, and it is hoped that the results of this study can have theoretical and practical significance on how to standardize online medical treatment and guide patients to seek medical treatment.

According to your suggestion, we have cited previous studies relevant to proposed hypotheses. We supplemented the following content: “People browse health information through the internet to make further medical decisions. On the one hand, the Internet provides diagnosis and treatment plans for almost all types of common diseases, including basic definition and symptoms, drug use methods, and contraindications [41-43]. Studies have shown that an increasing number of people tend to use the internet to obtain health care information [44-46], including older adults [47,48]. After the appearance of disease symptoms, patients will use the Internet to search for the type of disease corresponding to the symptoms and determine the severity of the disease [49,50]. If the patient judges that the disease is common and not serious, he will purchase drugs and self-diagnose according to the online treatment plan. The Internet offers many advantages for patients in comparison with the offline world, such as convenience, time saving, and reduced limitations on space and time. The statistical report on internet development has suggested that internet penetration has continued to grow and the popularity of the internet has gradually spread to the elderly from the young [51]. Based on the above analysis, this paper assumes:

Hypothesis 1: Internet use has a significant positive impact on self-treatment among older adults.

On the other hand, due to the multi-source of Internet information, it may aggravate the incompleteness of the individual knowledge of the disease and make individual more dependent on authoritative medical institutions [52]. In terms of rare diseases, it is difficult for different websites and platforms to provide patients with consistent information, and the uncertainty of diagnosis and treatment content may aggravate risk perception of patients’ disease. Patients often have access to information that could psychologically prepare the latter, but could also scare them. Online medical information focuses on universality and introductory content, and does not list the probability of potential consequences. As a result, patients often overestimate the negative consequences of rare diseases and increase the expected utility loss of misdiagnosis [53]. As far as serious diseases are concerned, the difficulty and individualization make patients have to undergo equipment inspections and professional diagnosis [54]. Community health service institutions located in common and chronic diseases face cross-border competition from Internet medical information. Top-level hospitals are in a professional monopoly position because they are good at handling difficult and complicated diseases [55,56]. Based on the above analysis, this paper assumes:

Hypothesis 2: When choosing medical facilities for common diseases, Internet use increases older adults's preference for top-level hospitals.

Hypothesis 3: When choosing medical facilities for serious diseases, Internet use increases older adults's preference for top-level hospitals..

  1. Methodology: Author did not develop its argument from appropriate theory. However, they explored models previously studied in the same area. However, the data is focused on elderly people. Therefore, the article represents secondary data from the content technique as the sample and methodology and relevant to present the context of the proposed theme. However, the empirical sample was fair enough to illustrate significant empirical results of this paper, and the methods employed are not appropriate. In addition, there are major concerns with sampling design and data nature. What is the population of banks (Sampling frame, total population, )?Why specifically were this period of 2011 2015 selected for this study?Why have China been chosen?What are the criteria for the selection of firms for data collection? (Sampling techniques, i.e., random, cluster, or judgmental sampling)?Regarding the methodology, more details and justification of why they did not employ panel data analysis?The author(s) did not define the data collection and sampling clearly.

Author response: Thank you for your preciseness. According to your suggestion, we have explained the sampling and data in detail. We supplemented the following content: “The data used in this study were from the China Health and Retirement Longitudinal Study (CHARLS), which a nationally representative survey of the population 45 years or above living in China, fu nded by Peking University (China), National Institute on Aging (China), and World Bank. Since 2011, CHARLS has conducted a survey every two years, sampling 28 (out of 31) provinces in China through multi-stage stratified Probability Proportionate to Size sampling (PPS), which can represent about 95% of China's population. The database is public, and more detailed description of the sampling design and process can be obtained from its website (http://charls.pku.edu.cn). In each survey wave, about 17,000 people living in 10,000 households in 150 counties/districts and 450 villages/resident committees (or villages) were surveyed by using the face-to-face computer-assisted personal interview. Due to the long-time span of follow-up survey, CHARLS research is faced with some temporary or permanent exits, which are offset by the new respondents, that is, data imbalance. The survey aims to provide a data base for population aging academic research and public health policy analysis by tracking and collecting a wide range of information on demographic and socio-economic characteristics, family relations and dynamics, wealth, employment, education, health status and functioning, biomarkers, health care and insurance [62-64]. The Institutional Review Board of Peking University granted ethical consent (IRB00001052-11015). To research the association between the Internet use and health seeking behavior and institution choice of the elder, we limited the samples to respondents who who fell ill in the last month. This paper selects the mixed cross-sectional data of the three phases of 2011, 2013 and 2015 as the analysis object. Compared with the cross-sectional data, the mixed cross-sectional data can increase the sample size, expand the sample representativeness, and obtain more precise estimates and more effective statistics. By eliminating the missing values, the final sample contains 9253 individuals.

  1. Results:The analysis is clearly provided to tie up with the findings. The author should provide a complete statistical analysis, i.e., skewness, and kurtosis in descriptive analysis, and correlation analysis. Results are poorly reported in the result section, and authors should discuss results with the help of statistical figures. There is a need for improvement in reporting results such as the author(s) should report [Beta value OR standard error (S.E) with significance level OR t-value; i.e., (β= xxx. P<0.01) OR (S. E= xxx, t > xxx). However, it could be more effective if the author(s) presents significant results with bold and asterisk (*). Why there is no diagnostic checks for panel data, i.e., VIF, auto correlation etc.

Author response: Thank you for your preciseness. According to your suggestion, we have improved the results report. The new results report is as follows: “Table 4 presents the results from the estimation of specification (1). In this paper, according to equations (1)-(3), logit estimates are made on the effect of medical options on using Internet, and the results are shown in Table 2. The results in column (1) show that Internet use has a significant positive impact on the self-treatment of common diseases (β=0.05, p<0.05). Specifically, using the Internet can increase the probability of the elderly self-treatment by 5%, so hypothesis 1 is supported. The results in column (2) show that Internet use has a significant positive impact on the elderly choosing top-level hospitals for treatment of common diseases (β=0.06, p<0.01), so hypothesis 2 is supported. However, the results in column (3) show that Internet use has no significant impact on the elderly choosing top-level hospitals for treatment of major disease (β=0.03, p>0.1). The hypothesis 3 is not supported.

Thank you for your preciseness. In fact, we have done a complete statistical analysis, but we have streamlined some non-essential tables for space.

  1. Discussion and findings:As results are clearly provided. However, there is no solid discussion on results. Author(s) should discuss the limitations of this study and future research direction in a constructive way. Hence, authors should write in prices and in a constructive way under a subsection of discussion and results. Author(s) did not discuss the theoretical and practical contribution of this study. Author(s) should discuss the theoretical and practical contribution of this study in the separate subsection under discussion for more clarity.

Author response: We deeply appreciate your thoughtful comments. According to your suggestion, we have rewritten the discussion section. We discussed the results in depth and discuss the limitations of this study and future research direction in a constructive way. The new discussion is as follows: “This paper investigated the impact of internet use on medical decisions medical decisions among Chinese older adults through several complementary method such ass logistic regression, propensity score matching (PSM) and fixed-effect models (FE) based on the 2011, 2013, and 2015 China Health and Retirement Longitudinal Study (CHARLS). The results showed that the internet had a certain effect on older adults’ health seeking behavior and institution choice. First, the elderly with Internet behavior are more inclined to self-cure when suffering from common diseases, especially for rural residents and middle-income groups, indicating that self-diagnosis and therapy can partially replace hospital care. Second, in terms of medical institution choices, the who use Internet were more inclined to choose top-level hospitals than community health service institution to treat common diseases. The study contains theoretical and practical consequences for how to govern internet health care and direct people to medical institutions, as well as a reference to internet medical treatment promotion and implementation.

Chinese older adults use Internet are more inclined to self-treatment than visiting hospitals, which is consistent with some research description. Yang et al pointed out that online medical platforms has become the “entrance” for many patients to see a doctor, which to a certain extent diverts the flow of patients with common diseases to high-level hospitals [72]. As the popularity of the internet has grown, surfing and selecting health information has become a standard procedure before deciding whether or not to visit a hospital [73]. The popularity of the Internet and mobile Internet has broken the medical information barriers, and the public can obtain diagnosis and treatment measures for common diseases from the Internet at low cost and conveniently. The emergence of online appointment registration services, online health care and monitoring, telemedicine, online diagnostic and treatment services, and medical supplies business related to medical services, medicines, online consultations enable people to enjoy online medical services more quickly, efficiently, and at low cost [74]. Overall, the internet has the potential to minimize barriers to common illness knowledge, to some extent lessen information asymmetry between patients and physicians, and to increase individuals' awareness and access to fundamental health knowledge, hence lowering the likelihood of utilizing medical services. Due to the current shortage of medical resources and the public’s thirst for medical resources, the integration of medical resources and the Internet is an important way to improve China’s lack of medical resources.

In contrast, this study discovered that the internet may increase the likelihood of seeking medical treatment from the top-level hospitals. The multi-source and uncertainty of medical information acquisition has exacerbated the inconsistency and incompleteness of individual’s cognition of diseases. Due to the limitation of professional knowledge, it is difficult for patients to identify the relevant information, and they are prone to be misled by the wrong medical information, which leads to health anxiety, for instance, physical symptoms are misinterpreted as signals of dangerous diseases, and there is a continual worry of being sick [75]. Ogasawara found that many websites offer harmful information about cancer, and the proportion of these websites is far higher than that of sites that offer reliable information about cancer treatment [76]. Brings “noise” and intensifies the “increasing tendency” of regular medical institutions. If the development of Internet medical care is allowed to develop savagely and irregularly, it may further aggravate the medical burden of high-level hospitals. The competent government departments should strengthen the supervision and guidance on the quality of medical information on the Internet to ensure the authority of medical information.

This study also has some limitations. First, although this paper has largely solved the endogenous problem caused by selection bias and missing variables that do not change over time through PSM and FE, but due to data limitations, no suitable instrumental variables have been found to further ensure the rigor of the results. Second, this is a survey on the middle-aged and elderly people in China. Because the health of the middle-aged and old people is quite different from other groups, all the conclusions of this study cannot be generalized to the general group. With the further development of online medical care, research on the mechanism of the influence of patients' medical behavior from the perspective of the Internet will be the direction of future research.

  1. Conclusion:Author should provide concluding statements rather that repetitive statements in the conclusion portion. Abstract and conclusion sections look similar, please revised conclusion section.

Author response: We deeply appreciate your thoughtful comments. According to your suggestion, we have rewritten the conclusion section. The new conclusion is as follows: “With the rapid development of a new generation of information technology, the dissemination and utilization of medical service information has been accelerated, and the functions of online medical services have also been continuously expanded. The internet has broken down the barriers to the knowledge of common diseases, shortened the gaps in health information accessibility, and has produced a slight substitution effect of self-diagnosis and treatment on hospital care. However, the knowledge monopoly of difficult and complicated diseases cannot be eliminated, and at the same time, the increase in inconsistent, incomplete, and commercialized medical information has also brought noise to decision making, and will blur the residents’ cognitive boundary of common diseases and severe diseases. Consequently, the rising tendency of visiting high-level medical institutions may be exacerbated, which is unable to guide patients to hierarchical diagnosis and treatment. The government should issue relevant policies to regulate the development of Internet medical care, guide patients to choose reasonable medical institutions based on their own conditions, so as to achieve the purpose of hierarchical diagnosis and treatment, save costs, and greatly improve Service efficiency and service quality.

  1. Citation and End References. The author did not cite the latest literature relevant to the target issue in this paper. Only a few articles are cited, which are published in the last five years. However, I strongly recommend more latest studies to cite and strengthen this paper.

Author response: Thank you for your preciseness. According to your suggestion, we have added a large number of the latest literature relevant to the target issue in this paper.

  1. Quality of Communication:The paper needs further proofreading. I suggest that more careful investigation of prior literature can make this paper distinguishable. Linking this article with prior studies does not seem to be sufficient, which weakens the justification of incremental contributions.

Author response: We deeply appreciate your thoughtful comments. According to your suggestion, we further proofread this paper.

Reviewer 2 Report

The manuscript presents the results of important analysis of possible relationships between internet use and medical-seeking behaviour among a large group of Chinese elder groups. However, there is one major problem and a number of minor problems.

Major problem.

The key variable is internet use. However, this is not specifically related to use of internet for information of medical information. Those using internet were younger, more urban residents, higher educated, higher income (although not tested for significance in Table 1. Are there other group characteristics that is the factor behind relationship between intern use and the three outcome variables? This need to be taken into consideration and properly be discussed.

Minor comments:

  1. Line 12: elder groups - need to be more specific in the abstract. The mean ages in table 1 is not  sufficient information. Median would give a more accurate information and the distribution. As we know that the internet use is highly dependent on age. In the end of the paper you used: middle-aged and elderly people.
  2. Line 38: Studies on Tertiary hospitals are discussed but the group in the analysis is all hospitals. 
  3. Line 80: Insert a subheading: Present study
  4. Line 114: It would be informative to present a concise the aims of the study here.
  1. Line 149: Insert “the perceived” in front of “medical and health needs”. It is not the professionally defined needs that is discussed.
  2. Line 153: what is meant by “cognitive deficiency”? Please rewrite this paragraph, which is difficult to understand. Maybe that preventive vs curative interventions should be discussed separately.
  3. Line 176: what is meant by “microscopic medical behaviors”?
  4. Line 181: Structural changes in the supply of medicinal information are likely to affect patients´ medical behavior. However, the authors do not know if the users of internet actually use internet as a source of medical information. This possible bias need to be discussed.
  5. Line 190. The paper 35 in the reference list do not refer to internet use.
  6. Line 248: mixed cross-sectional data: what is that? Do you use the llongitudinal data in any analysis?
  7. Line 278: It is not a very stringent argument for the validity of the proxy variable to be used.
  8. Line 293: Why is there no formal statistical tests applied to the data in Table 1.
  9. Line 303: It would be better to include the information on the three models already on line 295.
  10. Line 422: The low-income groups do they use internet at all? Moreover, the variables on medical seeking behavior do not differentiate where they actually seek hospital care. This part of the paper seems to partly contradict other results??
  11. Line 457: a reference would be needed for this information.
  12. Line 478: Is it really sure that it is the usage of Internet that has this significant effect: other candidates are type of residents, per capita income, gender, age…
  13. Line 512: aggravate the medical burden of high-level hospitals – but only hospitals are analysed in the paper.
  14. Please control that the papers referred to actual support your points. I have not had the possibility to cheque (see 9).

Author Response

  1. The key variable is internet use. However, this is not specifically related to use of internet for information of medical information. Those using internet were younger, more urban residents, higher educated, higher income (although not tested for significance in Table 1. Are there other group characteristics that is the factor behind relationship between intern use and the three outcome variables? This need to be taken into consideration and properly be discussed.

Author response: We deeply appreciate your thoughtful comments. Although the key variable in this study was not directly use of internet for information of medical information, many studies have shown that Internet use has a significant positive impact on medical knowledge and information. Takahash found that Internet use can improve individuals' health-related knowledge and attitudes. Patients demonstrated interest in online comparative health care information in Zwijnenberg et al's study. Because online browsing is a necessary condition for the overflow of Internet medical information, only individuals participating in online browsing activities have the opportunity to access Internet medical information. The process of online browsing greatly increases the possibility that individuals access medical information through the Internet. Especially for the elderly, they pay more attention to health information. In fact, a large number of studies have directly used Internet use as a proxy variable for information acquisition such as environmental protection, health and agricultural technique (Liu et al., 2021; Ma, Grafton, & Renwick, 2018; Fu & Akter, 2016; Hübler & Hartje, 2016; Tadesse & Bahiigwa, 2015). Therefore, it is reasonable to consider Internet use as a proxy variable for medical information access.

  1. Line 12: elder groups - need to be more specific in the abstract. The mean ages in table 1 is not  sufficient information. Median would give a more accurate information and the distribution. As we know that the internet use is highly dependent on age. In the end of the paper you used: middle-aged and elderly people.

Author response: Thank you for your preciseness. According to your suggestion, we have unified the research groups in the whole paper. This study focuses on older adults. Internet penetration has continued to grow and the popularity of the internet has gradually spread to the elderly from the young. Studying Internet medical information spillover on older adults in China is of particular interest given that China is the world's largest developing country and facing serious aging population which will inevitably further lead to a significant increase in the demand for medical services.

  1. Line 38: Studies on Tertiary hospitals are discussed but the group in the analysis is all hospitals.

Author response: Thank you for your preciseness and carefulness. The objective of this study is to explore health seeking behavior and institution choice under the background of the Internet era from the perspective of older adults, and to analyze whether Internet use leads patients to the community health service institution or hospital for medical treatment. In fact, the hospital here refers more to the higher level hospitals.

  1. Line 80: Insert a subheading: Present study

Author response: Thank you for your preciseness and carefulness. According to your suggestion, we have inserted “Present study”.

  1. Line 114: It would be informative to present a concise the aims of the study here.

Author response: We deeply appreciate your thoughtful comments. Considering that none of the articles published in this journal contains an independent literature review section, we integrate the literature review section into the introduction. According to your suggestion, we have defined the aims of this study in the introduction.

  1. Line 149: Insert “the perceived” in front of “medical and health needs”. It is not the professionally defined needs that is discussed.

Author response: We deeply appreciate your thoughtful comments. Considering that none of the articles published in this journal contains an independent literature review section, we integrate the literature review section into the introduction. Therefore, this subheading has been deleted.

  1. Line 153: what is meant by “cognitive deficiency”? Please rewrite this paragraph, which is difficult to understand. Maybe that preventive vs curative interventions should be discussed separately.

Author response: Thank you for your preciseness and carefulness. According to your suggestion, we have have removed the difficult phrase.

  1. Line 176: what is meant by “microscopic medical behaviors”?

Author response: Thank you for your preciseness and carefulness. According to your suggestion, we have have removed the difficult phrase.

  1. Line 181: Structural changes in the supply of medicinal information are likely to affect patients´ medical behavior. However, the authors do not know if the users of internet actually use internet as a source of medical information. This possible bias need to be discussed.

Author response: We deeply appreciate your thoughtful comments. In fact, many studies have shown that Internet use has a significant positive impact on medical knowledge and information. Takahash found that Internet use can improve individuals' health-related knowledge and attitudes. Patients demonstrated interest in online comparative health care information in Zwijnenberg et al's study. Because online browsing is a necessary condition for the overflow of Internet medical information, only individuals participating in online browsing activities have the opportunity to access Internet medical information. The process of online browsing greatly increases the possibility that individuals access medical information through the Internet.

  1. Line 190. The paper 35 in the reference list do not refer to internet use.

Author response: Thank you for your preciseness and carefulness. According to your suggestion, we have modified this reference.

  1. Line 248: mixed cross-sectional data: what is that? Do you use the llongitudinal data in any analysis?

Author response: The CHARLS conducted a follow-up survey on some interviewees, which can form a certain amount of panel data. However, to research the association between the Internet use and health seeking behavior and institution choice of the elder, we need to limit the samples to respondents who fell ill in the last month. This results in a significant reduction in sample data, which leads to inaccurate longitudinal analysis results. This paper selects the mixed cross-sectional data of the three phases of 2011, 2013 and 2015 as the analysis object. Compared with the cross-sectional data, the mixed cross-sectional data can increase the sample size, expand the sample representativeness, and obtain more precise estimates and more effective statistics. In addition, we also use the fixed effect model to test the robustness based on longitudinal data, and the results are still significant.

  1. Line 278: It is not a very stringent argument for the validity of the proxy variable to be used.

Author response: We deeply appreciate your thoughtful comments. we have modified this variable measurement. We supplemented the following content: “Like previous studies that consider Internet use through smartphones or mobile phones [65-68]. All respondents were asked whether they used a computer or mobile phone to surf the Internet. The value of the main treatment variable "Internet use" is 1 when respondents use the telephone or mobile phone to surf the Internet, otherwise it is 0. Because online browsing is a necessary condition for the overflow of Internet medical information, only individuals participating in online browsing activities have the opportunity to access Internet medical information. The process of online browsing greatly increases the possibility that individuals access medical information through the Internet. Therefore, the proxy variable is reasonable..”

Like previous studies that consider Internet use through smartphones or mobile phones [65-68]. All respondents were asked whether they used a computer or mobile phone to surf the Internet. The value of the main treatment variable "Internet use" is 1 when respondents use the telephone or mobile phone to surf the Internet, otherwise it is 0. Because online browsing is a necessary condition for the overflow of Internet medical information, only individuals participating in online browsing activities have the opportunity to access Internet medical information. The process of online browsing greatly increases the possibility that individuals access medical information through the Internet. Therefore, the proxy variable is reasonable. In fact, many studies have shown that Internet use has a significant positive impact on medical knowledge and information. Takahash found that Internet use can improve individuals' health-related knowledge and attitudes. Patients demonstrated interest in online comparative health care information in Zwijnenberg et al's study. Because online browsing is a necessary condition for the overflow of Internet medical information, only individuals participating in online browsing activities have the opportunity to access Internet medical information. The process of online browsing greatly increases the possibility that individuals access medical information through the Internet. The process of online browsing greatly increases the possibility that individuals access medical information through the Internet. Especially for the elderly, they pay more attention to health information. In fact, a large number of studies have directly used Internet use as a proxy variable for information acquisition such as environmental protection, health and agricultural technique (Liu et al., 2021; Ma, Grafton, & Renwick, 2018; Fu & Akter, 2016; Hübler & Hartje, 2016; Tadesse & Bahiigwa, 2015). Therefore, it is reasonable to consider Internet use as a proxy variable for medical information access.

  1. Line 293: Why is there no formal statistical tests applied to the data in Table 1.

Author response: We deeply appreciate your thoughtful comments. According to your suggestion, we have performed T-tests on the data in Table 1.

  1. Line 303: It would be better to include the information on the three models already on line 295.

Author response: Thank you for your preciseness and carefulness. According to your suggestion, we have added the meanings of all the characters in the formula.

  1. Line 422: The low-income groups do they use internet at all? Moreover, the variables on medical seeking behavior do not differentiate where they actually seek hospital care. This part of the paper seems to partly contradict other results??

Author response: We deeply appreciate your thoughtful comments. In fact, low-income groups rarely access the Internet, which may also be an important reason why Internet use no longer has an impact. However, due to the small sample size of low-income groups, it is difficult to conduct further research, which may also be an important problem to be solved in the future.

  1. Line 457: a reference would be needed for this information.

Author response: Thank you for your preciseness and carefulness. According to your suggestion, we have added references for this information.

  1. Line 478: Is it really sure that it is the usage of Internet that has this significant effect: other candidates are type of residents, per capita income, gender, age…

Author response: Thank you for your preciseness and carefulness. The regression model of this study controls a series of related variables and the PSM and FE model are adopted to solve the problem of selection bias and endogenous. The results obtained are robust.

  1. Line 512: aggravate the medical burden of high-level hospitals – but only hospitals are analysed in the paper.

Author response: Thank you for your preciseness and carefulness. According to your suggestion, we have changed “high-level hospitals” to “hospitals”.

  1. Please control that the papers referred to actual support your points. I have not had the possibility to cheque (see 9).

Author response: We deeply appreciate your thoughtful comments. According to your suggestion, we have added a number of the latest literature relevant to the target issue in this paper. We have cited previous studies to actual support the our points.

Round 2

Reviewer 1 Report

Dear Authors, 

Thank you for incorporating said comments and the paper is much improved.

Reviewer 2 Report

The manuscript is now very much improved.

The key variable internet use is now seen as a proxy and the discussion of the results is accordingly improved.

The mauscript is now ready for publishing